# DGTAT: DECOUPLED GRAPH TRIPLE ATTENTION NETWORKS

## ABSTRACT

The Message Passing Neural Network (MPNN) is a foundational paradigm in graph learning algorithms, demonstrating remarkable efficacy in its early implementations. Recent research has focused on using Transformer on graph data or combining Transformer with MPNNs to address issues like over-squeezing and over-smoothing while capturing long-range dependencies. However, Graph Transformers (GT) often perform poorly on small datasets. More seriously, much position and structure information encoded by GT-based methods is coupled with node attribute information, affecting node attribute encoding while propagating structure and position information, implicitly impacting on expressiveness. In this paper, we analyze the factors influencing the performance of graph learning models. Subsequently, we introduce a novel model, named DE-COUPLED GRAPH TRIPLE ATTENTION NETWORKS (DGTAT). Based on the MPNN+VN paradigm (Cai et al., 2023) and a sampling strategy, DGTAT effectively decouples local and global interactions, separates learnable positional, attribute, and structural encodings, and computes triple attention. This design allows DGTAT to capture long-range dependencies akin to Transformers while preserving the inductive bias of the graph topology. As a result, it exhibits robust performance across graphs of varying sizes, excelling on both large and small datasets. DGTAT achieves state-of-the-art empirical performance across a variety of node classification tasks, and through ablation experiments, we elucidate the importance of each decoupled design factor within the model. Compared to GT-based models, our model offers enhanced interpretability and flexibility.

## 1 INTRODUCTION

Graph machine learning is widely used in the domain of non-Euclidean data such as social networks, molecular structures, citation networks, etc. MPNN (Gilmer et al., 2017) is a classical paradigm for graph learning that uses graph topology as a strong inductive bias, and MPNN-based models such as GCN (Kipf & Welling, 2017), and GAT (Veličković et al., 2018) have achieved significant accomplishments in the early days. However, MPNN-based models suffer from the problems of over-smoothing (Li et al., 2018; Oono & Suzuki, 2021), over-squeezing (Alon & Yahav, 2021; Topping et al., 2022), and limited expressiveness (Xu et al., 2019; Loukas, 2020; Fountoulakis et al., 2023). In the propagation phase of MPNN, node attribute information is propagated along the graph topology. This approach propagates node attributes while expressing structure and position information, resulting in the coupling of attributes with structure and position information, which implicitly affects the expressiveness and interpretability of the model.

With the success of Transformer (Vaswani et al., 2023) in NLP (Devlin et al., 2019; Dai et al., 2019) and CV (Dosovitskiy et al., 2021; Liu et al., 2021), developing Transformers for graph data is attracting much interest (Dwivedi & Bresson, 2021). Transformers exhibit a remarkable proficiency in employing a global attention mechanism to apprehend long-range dependencies in graphs, which inherently alleviates the over-smoothing and over-squeezing. However, Transformer-based structures lose the topologically strong inductive bias of graphs (Dosovitskiy et al., 2021), require appropriate encodings to introduce structure and position information, have higher computational complexity, are more difficult to train and prone to be overfitting, and struggle to achieve good performance on small datasets. In addition, Transformers often incorporate structural or positional encoding into node or edge attributes (Kreuzer et al., 2021; Hussain et al., 2022), or introduce structure and posi-

Table 1: Encoding strategies of previous works. Encoding in the same cell indicates the coupling of information. It is worth noting that although the name LSPE includes both positional and structural encoding, the model uses one of RWSE or LapPE to initialize and learn, which is essentially structural encoding or positional encoding. On the other hand, GraphGPS presents a clear definition of both positional and structural encoding, but its model is based on MPNN+Transformer, which incorporates position and structure information into the node attributes for learning.

| | Positional Encoding(PE) | Structural Encoding(SE) | Attribute Encoding(AE) | Learnablity of PE/SE |
|---|---|---|---|---|
| SAN(Kreuzer et al., 2021) | Learned Laplacian encoding | | Transformer | Yes |
| LSPE(Dwivedi et al., 2022b) | Encoding initialized by LapPE or RRSE | | Transformer | Yes |
| Graph GPS | Based on MPNN+Transformer | | | Coupled with AE |
| SAT(Chen et al., 2022) | Based on Subgraph | | | Coupled with AE |
| NAGphormer(Chen et al., 2023) | Based on Subgraph | | | Coupled with AE |
| Graphormer(Ying et al., 2021) | Relative positional encoding | Centrality encoding | Transformer | No |
| GRIT(Ma et al., 2023) | RRWP + MLP | | Transformer | No |

Table 2: Interaction strategies of DGTAT

| | Global Interaction | Local Interaction |
|---|---|---|
| Position Information | Laplacian Encoding | RRWP + Jaccard Encoding |
| Structure Information | RWSE | Graph-topology based MPNN |
| Attribute Information | Virtual Node + Sampling | Graph-topology based MPNN |

tion information based on subgraph feature extraction (Chen et al., 2022; 2023), once again coupling attribute and structure information. Moreover, the global attention mechanism does not distinguish between neighboring nodes, coupling global message propagation with local message propagation. All of these couplings implicitly lead to computational and propagation redundancy, resulting in an implicit affect on expressiveness and interpretability.

A powerful graph learner should have advantages of both MPNN and Transformer, with the ability to capture global long-range dependencies while leveraging the inductive biases of graph topology and trying to avoid the pitfalls associated with global horizons. The properties it should have are:

1. Local and global sensing of node position information: Nodes should perceive their position in the whole graph, and locally they should perceive the relative positions of nearby nodes. The positional encoding should be learnable, sensing the importance of nearby nodes in real time.

2. Local and global sensing of node structure information: Nodes should perceive the graph topology in their neighborhood and transmit information based on the graph structure as an inductive bias, and the structural encoding should be learnable as well.

3. Local and global interactions of node information: The model should use the graph topology for local interactions of neighboring nodes while having global interaction capability to capture long-range dependencies.

It can be noticed that the above analysis classifies the influences on the performance of graph learners into six components: global position information, global structure information, global node attribute information; local position information, local structure information, and local attribute information. To the best of our knowledge, as shown in Table 1, many recent works have considered some of the above factors and achieved good results. However, there is a coupling in the information encoding and interaction, either structure/position information and node attribute are coupled to learn, or positional and structural encodings are unlearnable.

To avoid the potential redundancy and weakening of expressiveness caused by the coupling of each module, we adopt the strategy of Table 2, so that the structural encoding reflects only structure information, the positional encoding reflects only position information, and the modules are decoupled from each other. For position information, we use Laplacian positional encoding to introduce global position information of nodes. Relative Random Walk Probability (RRWP) is used on edges to introduce edge position information, and learnable Jaccard positional encoding is used on nodes

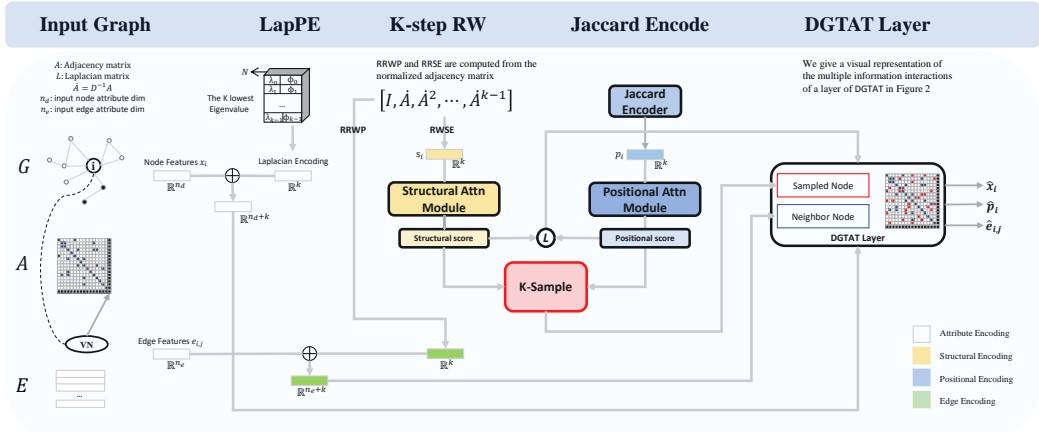

Figure 1: An abbreviated flowchart of DGTAT. $G$ represents an exampled graph, $E$ denotes the edge features of original graph, and $A$ denotes the adjacency matrix. The color filling scheme for $A$ is as follows: blue denotes connections in the original graph, black denotes connections to virtual nodes (linked to all nodes), and red denotes sampled connections. The sign $\oplus$ represents the splicing operation. The $L$ between positional score and structural score represents a learnable weight of our sampling strategy. Some normalization operations and MLP details are omitted for simplicity.

to introduce local relative position information, and through learning, nodes can perceive the importance of nodes around them. For structure information, we use MPNN as the basis to introduce global structural topology. Random Walk Structural Encoding (RWSE) is used as the initial structural encoding for each node to obtain pure structure information without position and attribute information to compute the structural similarity between nodes. For attribute information, inspired by MPNN+VN instead of Transformer (Cai et al., 2023), we introduce VN to introduce global attribute information interaction. Additionally, based on the results of structural and positional attention, we sample the K most relevant non-neighboring nodes at each layer for message passing to capture long-range dependencies beyond the limitations of the graph topology. The structure of our model is shown in Figure 1.

In summary, our primary contributions are as follows:

1. We provide a clear distinction between structure and position information. For the first time, we propose the idea of strictly decoupling the propagation of both position and structure information from the propagation of attribute features.

2. We present a design framework for graph learners with (global interaction, local interaction) * (position, structure, attribute) decoupled from each other, equipped with six decoupling elements. It reduces redundancy and potential impact between elements and is pluggable.

3. A feasible scheme is proposed based on the above design ideas, which encodes decoupled, pure and learnable positional and structural encodings for each node, adopts the basic framework of MPNN+VN, and has theoretically high expressiveness. DGTAT achieves state-of-the-art empirical performance across a variety of node classification tasks, and through ablation experiments, we illustrate that each decoupled design factor plays a role in the model.

## 2 RELATED WORKS

**MPNN+Virtural Node (VN).** Adding VNs to the graph structure was first proposed by (Gilmer et al., 2017). It connects all the nodes in the graph, which intuitively facilitates the nodes to obtain the global receptive field and improves the expressive power of the graph learner. Using VNs, the performance of some work has been improved (Hu et al., 2021a;b). Some theoretical studies on MPNN+VN have demonstrated that MPNN+VN-based paradigms can have expressive power comparable to GT (Cai et al., 2023).

**Graph Transformer (GT).** Considering the great success of Transformers in NLP and CV, there is great interest in extending transformers for graphs. Transformers excel in utilizing the global attention mechanism to capture the long-range dependencies in graphs, alleviating the problem of over-smoothing and over-squeezing, and achieving the SOTA effect on several large datasets. Currently, GT research is divided into two main directions, the first direction is to reduce the computational complexity and seek linear GT (Choromanski et al., 2022; Liu et al., 2022); the second direction is to design appropriate PE/SE (Kreuzer et al., 2021; Rampášek et al., 2023), or to combine MPNN with GT (Rampášek et al., 2023; Wu et al., 2022).

**Positional Encoding (PE) and Structural Encoding (SE).** The selection of PE/SE is an important factor that influences the performance of GT, and some previous work has proposed encodes that reflect shortest-path distances, identity-awareness, and spectral information (Li et al., 2020; You et al., 2021; Dwivedi et al., 2022a), or proposed learnable encodings (Dwivedi et al., 2022b). The categorization of PE and SE into local, global, and relative categories in GraphGPS (Rampášek et al., 2023) shares similarities with the conceptual framework presented in our paper, but the learnability and decoupling are not emphasized in practice. In conclusion, the PE/SE proposed in the above seminal work does not clearly delineate the boundary, and often contains both position and structure information, or couple with node attribute information while expressing position and structure information. This coupling of information can restrict the model's expressive power and may lead to incorrect results in certain cases, as demonstrated in Appendix B.

**Decoupled.** Decoupled representation learning (Bengio et al., 2014) endeavors to model the pivotal factors influencing the morphology of the data. Its objective is to ensure that a modification in a key factor exclusively induces a corresponding alteration in a specific feature, leaving other features unaffected. In the realm of graph learning, DAGNN (Liu et al., 2020) has made feature propagation and feature transformation independent through the idea of decoupling, which removes the potential impact of coupling propagation and transformation, allowing GCNs to train deeper layers. There are also many works on optimizing GNNs using the decoupling idea (Ma et al., 2019; Wang et al., 2020a;b). In this paper, we are inspired to decouple the attributes, structure information and position information of nodes with the aim of eliminating the potential expressiveness affect of coupling.

## 3 METHODS

In this section, we introduce our proposed DGTAT model, which decouples local and global interactions, separates learnable position, attribute, and structure information, and then computes triple attention, captures long-distance dependencies based on VNs as well as globally sampling according to structural and positional attention results. First, we introduce the selection of decoupling encodings and the corresponding initialization strategies (3.1 to 3.3). Then, we give the computation and learning strategies of DGTAT in 3.4. The flowchart of the whole model is shown in Figure 1, and an example of visualized information propagation details is shown in Figure 2.

We denote the constituents of the graph as G($V$, $\mathcal{E}$, $X$, $E$, $A$), where $V$ represents the set of nodes, $\mathcal{E}$ represents the set of edges, $X$ represents the node attribute feature matrix, $E$ represents the edge attribute feature matrix, and $A$ is the adjacency matrix. $x_i$ represents the attribute of node $i$ and $e_{ij}$ represents the edge attribute between node $i$ and $j$.

### 3.1 LEARNABLE DECOUPED PE

#### 3.1.1 LAPPE AS AN INITIALIZATION FOR NODE GLOBAL POSITIONAL ENCODING

Globally, each node should perceive its own global position. Laplacian encoding has been validated as an effective means to reflect the global position of a node without incorporating any attribute information. It is used in several works for the initialization of the positional encoding (Kreuzer et al., 2021). We select the eigenvectors corresponding to the lowest $k$ non-zero Laplacian matrix eigenvalues to splice with the node attribute features, not summed, as the initialized node features. $k$ is a hyperparameter chosen based on the size of the graph.

$$x_i = x_i \parallel p_i^{LapPE} \in \mathbb{R}^{d+k}$$

Here $x_i \in \mathbb{R}^d$ originally, where $d$ is the dimension of node attribute feature and $p_i^{LapPE} \in \mathbb{R}^k$ is the i-th row of the Laplace eigenvector matrix of the lowest K non-zero eigenvalues.

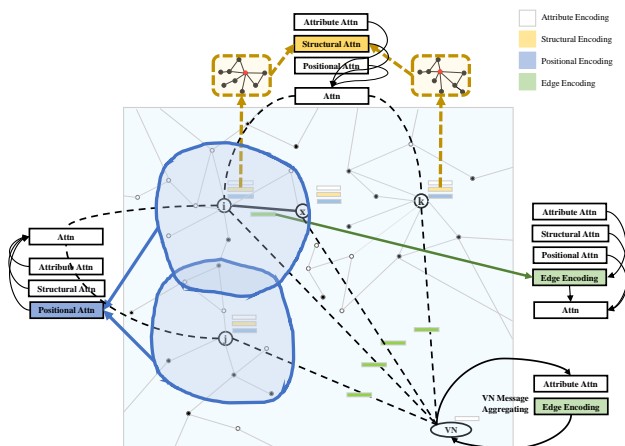

Figure 2: An example of visualized information propagation in DGTAT layers. This figure focuses on the information interaction of node $i$ with other nodes. It is divided into four parts, the left side of the figure represents the positional attention of node $i$ and node $j$ as the main factor activating the long-range communication, and the blue range around the node represents its visualized positional receptive field. The top of the figure represents the structural similarity of node $i$ and node $k$ as the main factor activating the long-range communication, and the yellow represents the visualized structure information from the structural encoding. The right and bottom right represent the message passing between node $i$ and its neighboring node $x$ and the communication with virtual nodes, respectively.

### 3.1.2 JACCARD PE AS AN INITIALIZATION FOR NODE LOCAL POSITIONAL ENCODING

Locally, a node should perceive its local position in the graph and perceive the associated node clusters. We initialize a Jaccard encoding for each node as a local positional encoding, which does not contain structure and node attribute information, but only the positional relationship of the node relative to other nodes, represented as $p_i = p_i^{Jaccard}$, and $p_{ik}$ denotes the k-th element of the positional encoding of node $i$. Nodes are able to sense the importance of nodes in their vicinity, which is used to compute weighted Jaccard similarity without message passing to capture positional relationships between non-neighboring nodes (Zhang et al., 2020). The positional encoding is learnable, it will also be learned at each layer, see Section 3.4. The computation of $p_i^{Jaccard}$ is shown in Appendix D.

### 3.1.3 RRWP AS AN INITIALIZATION FOR EDGE LOCAL POSITIONAL ENCODING

In addition to encoding position information for each node, we add positional encoding for each edge with the aim of further improving the local position sensing ability of the nodes in the graph as well as the virtual nodes. Virtual nodes have edges connecting all nodes, so this encoding improves the ability of virtual nodes to capture global long-range dependencies, which is the key to improving the performance of large graphs.

Drawing from prior work, we employ RRWP as the positional encoding for edges (Li et al., 2020; Mialon et al., 2021). Previous research has demonstrated that with appropriate architectures, RRWP is more expressive than shortest path distances (Ma et al., 2023) and can capture large graph propagation matrices. Specifically, we use K-step RRWP encoding and edge attribute features for splicing as initialized edge features. The edge encoding is also learnable, with update formulas described in Section 3.4, which enables real-time adjustments in the attention levels of virtual nodes towards individual nodes, as well as the attention levels of individual nodes towards their neighboring counterparts.

Let $D$ be the diagonal degree matrix. Define $\tilde{A} = D^{-1}A$, and note that $\tilde{A}_{i,j}$ is the probability that $i$ hops to $j$ in one step of a simple random walk. The proposed RRWP initial positional encoding is defined as follows:

$$p_{i,j}^{RRWP} = [I, \tilde{A}, \tilde{A}^2, \cdots \tilde{A}^{K-1}]_{i,j} \in \mathbb{R}^k \qquad (1)$$

$$e_{ij} = e_{ij} \parallel p_{i,j}^{RRWP} \in \mathbb{R}^{d_e+k}$$

Here $e_{ij} \in \mathbb{R}^{d_e}$ originally, where $d_e$ is the dimension of edge attribute feature, and $p_{i,j}^{RRWP} \in \mathbb{R}^k$ is the probability that $i$ hops to $j$ in zero to $k-1$ step of random walks. $\parallel$ represents the concat operation. $k$ is a hyperparameter chosen based on the size of the graph.

## 3.2 LEARNABLE DECOUPED SE

### 3.2.1 RWSE AS AN INITIALIZATION FOR NODE GLOBAL STRUCTURAL ENCODING

Unlike positional encoding, node structure information allows nodes to perceive the subgraph topology in their neighborhood, and to perceive the cluster structure of the surrounding nodes without knowing the specific nodes around them, independent of the position. Many previous works do not distinguish well between node position and structure information, often coupling position and structure information or interacting structure information with the flow of node attributes on the graph, which affects the expressiveness of graph learners.

The RWSE applied by (Dwivedi et al., 2022b) is proposed as a positional or structural encoding and has proven effective by previous works. Encoded by the probability of arriving at the node itself by a 1-k step wandering, it does not contain any attribute information or position information of the node, reflecting purely structure information, which includes the degree information of the node and the graph topology information of the K-step receptive field. By calculating structural attention, similar subgraph topologies at any position can interact. Thus, similar molecular cluster structure centers can interact with information. We provide a learnable matrix for structure encoding.

$$s_i^{RRSE} = p_{i,i}^{RRWP} \in \mathbb{R}^k$$

Similar to Equation 1, the $s_i^{RRSE} \in \mathbb{R}^k$ is the probability that $i$ hops to itself in zero to $k-1$ step of random walks.

### 3.2.2 INTERACTION OF LOCAL STRUCTURE INFORMATION VIA GRAPH TOPOLOGY BASED MPNN STRUCTURES

For local structure information, the MPNN-based structure leverages the neighborhood of each node for interaction. The aggregation and propagation of arbitrary information with neighboring nodes essentially tap into the strong inductive bias of the graph topology. This enables the effective utilization of the node's local structure information.

## 3.3 DECOUPLED NODE ATTRIBUTE ENCODING

### 3.3.1 GLOBAL ATTRIBUTE INTERACTION VIA VIRTUAL NODES

We use the MPNN+VN structure for the global interaction of node attributes. The MPNN+VN structure with O(1) depth and O(1) width approximates a linear Transformer representation while O(1) depth and O($n^d$) width can approximate the full Transformer (Cai et al., 2023). Here we adopt the structure with width K, whose expressiveness is in between the above two. Through the appropriate VN node interaction and update strategy, it extends the attribute information interaction between nodes to the global scale, improves the interaction capability between long-range nodes, and leaps out of the limitation of the traditional MPNN structure. The update formulas of VN nodes are shown in 3.4.

### 3.3.2 GLOBAL INTERACTIONS SAMPLED FROM STRUCTURAL AND POSITIONAL ATTENTION RESULTS

For each pair of non-virtual nodes, we compute structural attention and positional attention based on structural encoding and positional encoding. Nodes exhibiting high structural similarity or strong positional correlation are sampled for interaction. Here we select the K most relevant nodes per layer according to the algorithm to connect and participate in message aggregation and propagation in this layer. $K$ is a hyperparameter chosen based on the size of the graph.

As in 3.2.2, the flow of node attributes along the graph topology naturally facilitates the local interaction of node attributes. We next detail our model DGTAT in 3.4.

## 3.4 DGTAT

In this section, we outline the specific computations carried out by the DGTAT model. For simplicity, we omit the normalization operation, the BatchNorm operation and the bias term we used in the experiment from the description.

For each node, we classify node interactions into local and global interactions, and global interactions encompass interactions with both sampled nodes and virtual nodes. After initialization, for each node, we first compute the structural and positional attention between pairs of nodes:

$$s_{i,j} = a_s^T LeakyReLU(W_s[s_i \parallel s_j]) \tag{2}$$

$$p_{i,j} = J(p_i, p_j) = \frac{\sum_{k \in (V_i \cup V_j)} \min(p_{ik}, p_{jk})}{\sum_{k \in (V_i \cup V_j)} \max(p_{ik}, p_{jk})} \tag{3}$$

Here we adopt the GATv2(Brody et al., 2022) style to compute the structural attention $s_{i,j}$, where $a_s^T \in \mathbb{R}^{d'}$ and $W_s \in \mathbb{R}^{d' \times 2k}$ are learnable weight matrices. $J(.,.)$ is the Jaccard similarity algorithm to comupte the positional attention $p_{i,j}$, and $V_i$ denotes the neighborhood of node $i$.

For the neighboring nodes of the node, we update the edge encoding based on the structural attention and node attributes. In this phase, the edge encoding for the neighboring nodes incorporates the node attribute information and subsequently the information interaction weights of the neighboring nodes are calculated based on the positional attention, structural attention and edge encoding of the neighboring nodes.

$$\hat{e}_{ij} = \sigma(\rho(W_{AE}[x_i \parallel x_j] \odot W_{EW} e_{ij}) + s_{ij} W_{Eb} e_{ij}) \tag{4}$$

$$a_{ij} = \alpha s_{ij} + \beta p_{ij} + W_A \hat{e}_{ij} \tag{5}$$

$\sigma$ is a non-linear activation ($ReLU$ by default), $W_{AE} \in \mathbb{R}^{d' \times 2d}$, $W_{Eb} \in \mathbb{R}^{d' \times d_e}$ and $W_A \in \mathbb{R}^{1 \times d'}$ are learnable weight matrices. $\odot$ indicates elementwise multiplication, and $\rho$ is the signed-square-root which stabilizes training by reducing the magnitude of large inputs.

For the non-neighbor nodes of the node, we sample the nodes based on the structural attention and positional attention calculations. We select the K most relevant nodes according to the algorithm and calculate the information interaction weights of the sampled virtual edges.

$$a_{ij} = \alpha s_{ij} + \beta p_{ij} + \rho(a_e^T LeakyReLU(W_{AE}[x_i \parallel x_j])) \tag{6}$$

$\alpha$ and $\beta$ are learnable parameters. $\parallel$ represents the concat operation, and $\rho$ is the signed-square-root operation. $j \in K_i$, where $K_i$ denotes the sampled sets of node $i$.

We then perform node attribute updates and positional encoding updates. Before this, we perform a softmax or linear normalization of the attention matrix.

$$\hat{x}_i = \sum_{j \in (V_i \cup K_i)} a_{ij} W_V x_j \tag{7}$$

$$\hat{p}_i = \sigma(p_i + \rho(W_P a_i)) \tag{8}$$

Here $K_i$ denotes the sampled sets of node $i$. $W_V \in \mathbb{R}^{d \times d'}$ and $W_P \in \mathbb{R}^{n \times n}$ are learnable weight matrices.

For virtual nodes, we adopt the following update strategy, noting that virtual nodes are neighbor nodes of all graph nodes.

$$\hat{x}_{vn} = \gamma_{vn}(x_{vn}, \phi_{j \in [n]}(x_{vn}, x_j, e_{vn,j})) \tag{9}$$

Here $\gamma$ is a $\mathbb{R}^d \times \mathbb{R}^{d'} \longrightarrow \mathbb{R}^d$ update function and $\phi$ is a $\mathbb{R}^d \times \mathbb{R}^d \times \mathbb{R}^{d_e} \longrightarrow \mathbb{R}^{d'}$ message function.

## 4 EXPERIMENTS

### 4.1 EXPERIMENTAL SETTINGS

**Datasets.** We evaluated our model on 8 node classification benchmark datasets, among which 3 are homophilic, 3 are heterophilic and 2 are relatively large-scale datasets. In all the experiments, we use the publicly available train-validation-test splits. Detailed information is provided in A.

Table 3: Node classification performance on homophilic graphs and heterophilic graphs. The best results are highlighted by a green background. OOM indicates the out-of-memory error.

| Type | Homophilic graphs | | | Heterophilic graphs | | |
|---|---|---|---|---|---|---|
| Dataset | Pubmed | Citeseer | Cora | Texas | Cornell | Actor |
| GCN-II | 80.06 ± 0.7 | 73.20 ± 0.8 | 85.24 ± 0.6 | 78.45 ± 6.7 | 77.32 ± 7.5 | 35.57 ± 0.8 |
| GAT | 78.96 ± 0.4 | 71.80 ± 0.6 | 83.76 ± 0.5 | 60.44 ± 6.0 | 58.22 ± 4.0 | 30.28 ± 0.9 |
| GATv2 | 79.22 ± 0.3 | 72.10 ± 1.0 | 83.95 ± 0.5 | 60.28 ± 7.0 | 58.41 ± 3.9 | 30.13 ± 0.9 |
| GT | 79.08 ± 0.4 | 70.10 ± 0.8 | 82.22 ± 0.6 | 84.18 ± 5.4 | 80.16 ± 5.1 | 36.17 ± 0.8 |
| Graphormer | OOM | 72.40 ± 0.7 | 82.31 ± 0.6 | 84.27 ± 5.8 | 80.00 ± 4.9 | OOM |
| GraphGPS | 79.94 ± 0.3 | 72.40 ± 0.6 | 82.44 ± 0.6 | 82.21 ± 6.9 | 80.06 ± 5.1 | 35.85 ± 0.9 |
| NAGphormer | 80.57 ± 0.3 | 72.80 ± 0.8 | 84.20 ± 0.5 | 80.12 ± 5.5 | 79.89 ± 7.1 | 36.37 ± 1.0 |
| DGTAT (ours) | 80.71 ± 0.4 | 73.40 ± 0.6 | 85.12 ± 0.4 | 84.39 ± 5.1 | 82.14 ± 6.0 | 36.43 ± 0.9 |

Table 4: Node classification performance on large graphs.

| Dataset | Aminer-CS | Amazon2M |
|---|---|---|
| GraphSAINT(Zeng et al., 2020) | 51.91 ± 0.20 | 75.20 ± 0.18 |
| GRAND+(Feng et al., 2022) | 54.76 ± 0.23 | 75.83 ± 0.21 |
| NAGphormer | 56.21 ± 0.42 | 77.43 ± 0.24 |
| DGTAT (ours) | 56.38 ± 0.51 | 78.49 ± 0.29 |

**Baselines.** We compare DGTAT with 7 advanced baselines, including various MPNN-based GNNs, such as GCNII (Chen et al., 2020), GAT, GATv2, and GT-based methods such as GT, Graphormer, GraphGPS, and NAGphormer. Some models do not report evaluation results for the node classification task in their paper, or evaluate dataset segmentation in a way that is inconsistent with publicly available methods. For these models, we reproduce them and test their performance based on local replication. Our comparison results are shown in Table 3 and Table 4.

**Evaluation details.** We use the Adam optimizer (Kingma & Ba, 2017) to train the models and select the best parameters based on early stopping. When measuring model performance, we use random seeds and report the mean ± standard deviation (SD) of classification accuracy over 50 trials.

## 4.2 EXPERIMENTAL RESULTS

On heterophilic graphs, where node classification requires high structural and positional awareness, methods such as attribute-attention-based GNNs perform poorly, while models with structural and positional encodings perform better. Compared to baselines, DGTAT achieves the best performance on all tested datasets, both on smaller graphs (Texas, Cornell) and relatively larger graph (Actor).

On homophilic graphs, the relative importance of each neighbor does not vary as much as on heterophilic graphs, so attention-based GNNs perform well. In addition, with a small number of train labels, deep and complex models are prone to overfitting, leading to performance degradation. Nevertheless, our model still shows strong performance, ranking first in 2 of the 3 homophilic datasets.

To evaluate the performance of DGTAT on large datasets and its ability to capture long-range dependencies, we conducted experiments on two relatively large datasets. Most existing graph transformers cannot work on such large datasets due to their high computational cost, while MPNN+VN-based DGTAT has good scalability. We compare our model with 2 scalable GNNs and NAGphormer. As shown in Table 4, DGTAT achieves the best performance on all tested datasets, indicating its good ability to work on large datasets.

## 4.3 ABLATION EXPERIMENTS

We conducted elaborate ablation experiments on the Citesser, Cora, and Pumbed datasets. We start with a model with only attribute attention as a baseline. We first add learnable decoupled structural encoding and the sampling strategy. Subsequently, we incorporate learnable decoupled positional

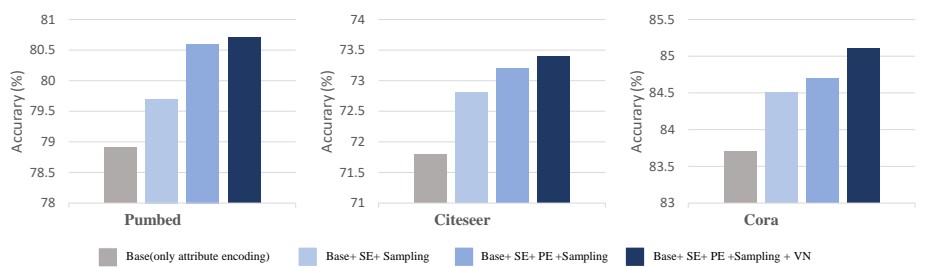

Figure 3: The results of ablation experiments

Table 5: The results with or without learnability.

|  | Pubmed | Citeseer | Cora | Texas | Cornell | Actor | AMiner-CS | Amazon2M |
|---|---|---|---|---|---|---|---|---|
| No-learnability | 80.03 | 72.60 | 84.37 | 84.17 | 80.21 | 36.29 | 56.17 | 76.84 |
| Learnable | 80.71 | 73.40 | 85.12 | 84.39 | 82.14 | 36.43 | 56.38 | 78.49 |
| Gain | +0.68 | +0.80 | +0.75 | +0.22 | +1.93 | +0.14 | +0.21 | +1.65 |

encoding. Finally, we add virtual nodes. The experimental results are as Figure 3. Each design module improves the accuracy of the model on different datasets.

It's worth noting that when decoupled structural encoding and sampling strategy are added first, they produce a more noticeable enhancement in performance. Conversely, when the order of addition is reversed, with positional encoding introduced initially, it results in a bigger improvement as well. This phenomenon arises from the fact that certain nodes can be distinguished by both structure and position information, leading to an overlap.

For the addition of virtual nodes, the extent of performance improvement varied depending on the dataset, suggesting that the significance of long-range dependencies differs between datasets. These results from the ablation experiments justify our analysis and improve the interpretability of the model.

On the other hand, We compare DGTAT to its variant whose structural and positional encoding are not learnable. The results are shown in Table 5, indicating that the learnability of PE and SE can improve the model performance for the node classification task. We can observe that the gains of learnability vary in different datasets since different graphs have varying sensitivity to real-time sensing of structure and position.

## 5   CONCLUSION

We provide a clear distinction between structural and positional encoding in graph learning, share the idea of strictly decoupling the propagation of position/structure information from the propagation of attributes, and analyze the factors that a powerful graph learner should possess. We propose DGTAT, a novel and powerful graph learner based on MPNN+VN and sampling. By computing attention through learnable structural and positional encoding and sampling relevant nodes, together with virtual nodes, DGTAT effectively captures the long-range dependency and maintains the inductive bias based on graph topology, allowing for robust performance on both large and small graphs. Our ablation experiments illustrate the good interpretability of our model.

In future work, we will extend DGTAT to graph-level classification tasks. Leveraging the attributes of virtual nodes, we are capable of extracting comprehensive attribute information for the entire graph, which is intuitively superior to the global node pooling readout strategies commonly used in graph classification, such as mean, max, and sum, etc.

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

## A  MORE EXPERIMENT DETAILS

This appendix provides the specific sources of our dataset, the specific basis for its splitting, and additional experiments for the choice of the hyperparameter K. All of the code is available at `https://github.com/DGTAT/DGTAT`.

**Details of Datasets**

The Pubmed, Citeseer, and Cora datasets are citation networks (Yang et al., 2016). Each node represents a research article and two nodes are adjacent if there is a citation between two articles. The node attributes are the bag-of-words features, and the node label is the category of the research domain of the article.

The Texas and Cornell datasets are extracted from the WebKB dataset (Pei et al., 2020). Each node represents a webpage and two nodes are adjacent if there is a hyperlink between the two webpages. The node attributes are the bag-of-words features, and the node label is the category of the webpage.

The actor dataset is the actor-only induced subgraph of a film-director-actor-writer network obtained from Wikipedia webpages (Pei et al., 2020). Each node represents an actor and two nodes are adjacent if the two corresponding actors appear on the same Wikipedia webpage. The node features are derived from the keywords on the Wikipedia webpage of the corresponding actor, and the node label is determined by the words on the webpage.

AMiner-CS (Feng et al., 2021) are citation networks in which nodes represent papers and edges represent citations. Amazon2M (Chiang et al., 2019) are co-purchase networks, where nodes indicate goods and edges indicate that the two connected goods are frequently bought together.

**Train/Val/Test Split**

In all of the datasets, we used publicly available splits. The public splits of Cora, Citeseer, and Pubmed are given by (Yang et al., 2016). The public splits of the heterophilic graphs are provided by (Pei et al., 2020). For large graphs, we refer to works of (Feng et al., 2021) and (Chiang et al., 2019). More details are shown in Table 6.

**Details of hyperparameter**

We perform hyperparameters for each baseline referring to the official implementations. For the DGTAT, we try layers in $\{1, 2, ..., 5\}$ and hidden dimension in $\{64, 128, 256\}$. We use a learning rate of in $\{1e-3, 5e-4, 1e-4\}$, weight decay of $\{1e-4, 5e-5, 1e-5\}$ and the dropout rate in $\{0.1, 0.3, 0.6\}$.

**More experiments for K**

We conducted experiments on 3 small datasets and 2 large datasets for the hyperparameter K. For simplicity, K is applied uniformly to denote the number of steps in the random walk, the count of virtual nodes, and the number of samples per node. Intuitively, as the size of the graph increases, a larger value of K is needed to achieve optimal performance. The results of our experiments are depicted in the figure 4. The value of K required to achieve optimal performance on a small dataset is approximately the number of nodes/1000, while the two large graphs achieve excellent performance at K = 10 and K = 15, respectively.

**More ablation experiments**

We have designed more detailed ablation experiments: Exp1 illustrates the contribution of our sampling strategy; Exp2 and 3 further support the importance of the decoupling strategy; Exp4 illustrates the crucial role of our design framework and shows that the performance improvement is not due to a simple stacking of encodings. Specifically, as follows:

1: Remove DGTAT's sampling strategy (other computations are consistent with standard DGTAT).

As shown in table 7, the experimental results show a significant decrease in accuracy for all datasets tested, especially for large-scale graphs, further illustrating the ability of the sampling strategy to capture long-range dependencies.

2: Couple the structural encoding and positional encoding in the DGTAT framework.

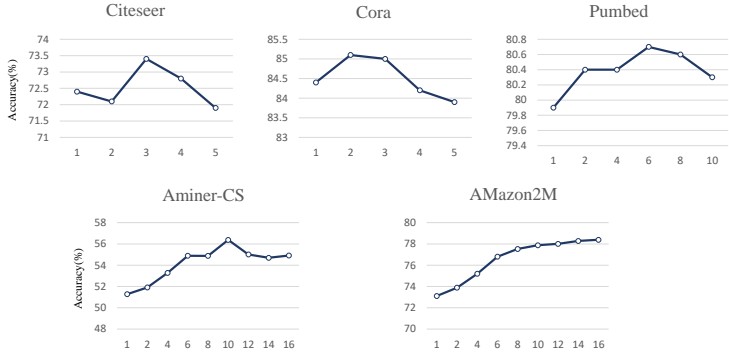

Figure 4: The results of experiments for the hyperparameter K

Table 6: The details of datasets

|  | Pubmed | Citeseer | Cora | Texas | Cornell | Actor | AMiner-CS | Amazon2M |
|---|---|---|---|---|---|---|---|---|
| Nodes | 19,717 | 3,327 | 2,708 | 183 | 183 | 7,600 | 593,486 | 2,449,029 |
| Edges | 44,324 | 4,552 | 5,278 | 279 | 277 | 26,659 | 6,217,004 | 61,859,140 |
| Features | 500 | 3,703 | 1,433 | 1,703 | 1,703 | 932 | 100 | 100 |
| Classes | 3 | 6 | 7 | 5 | 5 | 5 | 18 | 47 |
| Spilt(%) | 0.3/2.5/5.0 | 5.2/18/37 | 3.6/15/30 | 48/32/20 | 48/32/20 | 48/32/20 | (Pei et al., 2020; Chiang et al., 2019) | |

Unlike standard DGTAT, this experiment splices or sums the normalized structural encoding and positional encoding into a coupled encoding. Then we use the coupled encoding to compute global attention, and then sample long-range dependency based on the attention results. As shown in table 8, experimental results show significantly lower performance than standard DGTAT, illustrating the positive contribution of the decoupling of positional encoding and structural encoding.

3: Couple the structural encoding and positional encoding into the attribute features in the DGTAT framework.

We use this structure, position, and attribute coupled encoding to compute global attention, and then sample based on the results.As shown in table 8, a more pronounced decrease in performance occurs, illustrating the positive contribution of the decoupling of attribute encoding and other encodings.

4: In some baseline models of MPNN and GT (specifically, GATv2 and GPS), we directly incorporate the different encodings used in our work.

As shown in table 9, With the addition and coupling of encodings, the performance does not improve or improves only weakly. Even the direct incorporation of these coupled encodings instead leads to a severe decrease in accuracy. Comparing the ablation experiment in section 5, as shown in Figure 3, performance steadily improves as the same encoding is progressively added to our DGTAT framework. This experimental result illustrates that leaving the decoupling framework of DGTAT, simple stacking of encodings cannot improve GT performance, further supporting that our idea of decoupling makes a contribution and our framework is solid.

Table 7: The results with or without sampling strategy.

|  | Pubmed | Citeseer | Cora | Texas | Cornell | Actor | AMiner-CS | Amazon2M |
|---|---|---|---|---|---|---|---|---|
| DGTAT without sampling | 79.88 | 73.1 | 83.97 | 79.15 | 78.24 | 35.77 | 54.97 | 76.51 |
| DGTAT | 80.71 | 73.4 | 85.12 | 84.39 | 82.14 | 36.43 | 56.38 | 78.49 |
| Gain | +0.83 | +0.3 | +1.15 | +5.24 | +3.9 | 0.66 | +1.41 | +1.98 |

Table 8: The results of ablation experiments for supporting the role of decoupling.

|  | Pubmed | Citeseer | Cora |
|---|---|---|---|
| DGTAT | 80.71 | 73.4 | 85.12 |
| DGTAT with coupled SE+PE | 79.94 $(-0.75)$ | 72.4 $(-1.0)$ | 82.44 $(-2.68)$ |
| DGTAT with coupled SE+PE+AE | 78.96 $(-1.75)$ | 71.8 $(-1.6)$ | 82.76 $(-2.36)$ |

Table 9: The results of ablation experiments for supporting the role of our decoupled framework.

|  | Pubmed | Citeseer | Cora |
|---|---|---|---|
| GATv2 | 79.22 | 72.1 | 83.95 |
| GATv2+LapPE | 79.27 $(+0.05)$ | 72.3 $(+0.2)$ | 84.21 $(+0.26)$ |
| GATv2+LapPE+RRSE | 78.96 $(-0.26)$ | 71.9 $(-0.2)$ | 83.72 $(-0.23)$ |
| GATv2+LapPE+RRWP+RRSE | 77.61 $(-1.61)$ | 69.8 $(-2.3)$ | 80.53 $(-3.42)$ |
| GraphGPS (without PE/SE) | 79.58 | 72.1 | 82.26 |
| GraphGPS+LapPE | 79.94 $(+0.36)$ | 72.4 $(+0.3)$ | 82.44 $(+0.18)$ |
| GraphGPS+LapPE+RRSE | 80.53 $(+0.95)$ | 72.9 $(+0.8)$ | 83.29 $(+1.03)$ |
| GraphGPS+LapPE+RRWP+RRSE | 79.09 $(-0.49)$ | 71.9 $(-0.5)$ | 82.57 $(+0.31)$ |

## B   WHY WE NEED DECOUPLING IN GRAPHS

**1-Weisfeiler-Leman test**

The 1-WL test is a node-coloring algorithm within the hierarchy of Weisfeiler-Leman (WL) heuristics for graph isomorphism (Leman, 2018). It operates through iterative updates of node colors based on their 1-hop local neighborhood until a point is reached where further iterations no longer alter the node colors.

**Expressive power of MPNNs and PE/SE**

The constraints of MPNNs in discerning non-isomorphic graphs were first rigorously examined in the works of (Xu et al., 2019; Morris et al., 2021). This analysis stems from the established equivalence between MPNNs and the 1-WL isomorphism test, as presented by (Leman, 2018) in 1968. Consequently, MPNNs may exhibit poor performance on graphs featuring multiple symmetries within their original structure, encompassing node and edge isomorphisms (Murphy et al., 2019; Srinivasan & Ribeiro, 2020).

A natural approach to improve the expressivity power of the 1-WL test is to examine higher-order interactions between nodes with k-tuple of nodes with $k >= 3$. To this end, works that are more powerful than 1-WL utilize higher-order graph neural networks or Transformer with PE/SE, and there are theoretical proofs that using PEs such as LapPE and RRWP or SE such as RRSE can lead to a more powerful expressivity power than 1-WL test. We will not repeat the proof here, for details refer (Rampášek et al., 2023; Dwivedi et al., 2022b).

**Expressiveness analysis of DGTAT**

In the DGTAT architecture, we use decoupled structural and positional encodings, both of which perceive the $k >= 3$ node information as well as gain more expressive power than the 1-WL test through the global positional structural attention mechanism. So DGTAT has excellent performance on graphs featuring multiple symmetries within their original structure, encompassing node and edge isomorphisms, and can distinguish the CSL graphs which cannot be learned by 1-WL or MPNNs. Further, compared to the GT with SE/PE, through the decoupling of PE, SE and AE, DGTAT can distinguish some graphs with more sensitive position, structure and attribute information that coupled PS/SE cannot learn.

Here we provide additional visualization examples of the molecular graph propagation process to illustrate why the decoupling of positional, structural and attribute encodings is necessary. Through long-range communication based on sampling and the addition of virtual nodes, the DGTAT layer is

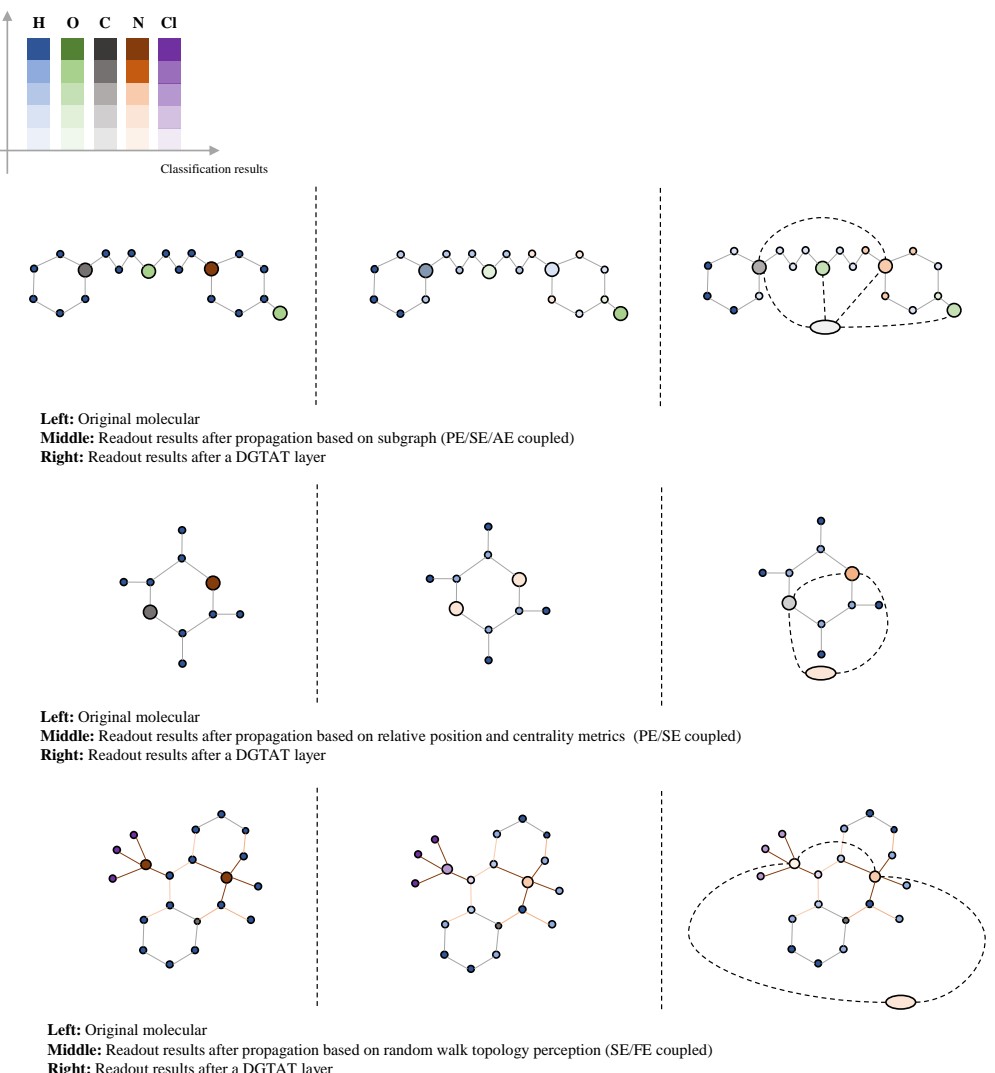

Figure 5: Some visualization examples of the molecular graph propagation process

usually able to produce correct results in cases where the coupled encoding leads to incorrect results. Specific instructions are shown in Figure 5.

## C    DGTAT'S ABILITY TO AVOID OVER-SMOOTHING

In this section, we provide a theoretical and experimental analysis of the over-smoothing problem of DGTAT. We introduce a quantitative metric of smoothness. We theoretically and experimentally illustrate the ability of DGTAT to perceive structural and positional information while modifying less node attribute information, which provides a stronger resistance to over-smoothing compared to GT, resulting in a better performance with fewer layers to perceive the critical long-range dependencies.

We use the average Euclidean distance of the node attributes as a measure of smoothness and calculate the smoothness at each layer, with a larger decline indicating that the model is less resistant to over-smoothing. We proved theoretically that DGTAT performs better than GT in resisting over-smoothing. We also conducted experiments in several datasets and found that DGTAT outperforms

GT in our smoothness metric, shown in figure 6. Our quantitative metric of smoothness is as follows:

$$
\begin{aligned}
sm(G) &= \frac{1}{n} \sum_{i \in V} \frac{1}{n-1} \sum_{j \in (V-i)} D(x_i, x_j) \\
&= \frac{1}{n} \sum_{i \in V} \frac{1}{n-1} \sum_{j \in V} \frac{1}{2} \left\| \frac{x_i}{\| x_i \|} - \frac{x_j}{\| x_j \|} \right\| \\
&= \frac{1}{2n(n-1)} \sum_i \sum_j \left\| \frac{x_i}{\| x_i \|} - \frac{x_j}{\| x_j \|} \right\|
\end{aligned}
$$

Here $\| \, . \, \|$ denotes the Euclidean norm. $sm(G)$ is negatively related to the overall smoothness of node attributes in graph G. In the following, we prove theoretically that DGTAT is further away from over-smoothing than GT when sampling long-range dependencies of equal attention. First, we give the overall node update formula for DGTAT, where $N(i)$ is the set of neighboring nodes and $K(i)$ is the set of sampled nodes. For simplicity, we omit the details of the model here and use only $\Psi, \varphi, \psi, \phi$ to represent the learnable aggregation and update methods. Ultimately, our node update can be written as $X A_{Ni} + X A_{Si}$, where $A_N$ only pays attention to neighboring nodes and $A_S$ only pays attention to sampled nodes.

$$
\begin{aligned}
\hat{x}_i &= \sum_{j \in N(i)} \Psi(\varphi(s_i, s_j), \psi(p_i, p_j), \phi(x_i, x_j, e_{i,j})) x_j + \sum_{j \in K(i)} \Psi(\varphi(s_i, s_j), \psi(p_i, p_j), \phi(x_i, x_j)) x_j \\
&= \sum_{j \in N(i)} [\alpha s_{ij} + \beta p_{ij} + \phi(x_i, x_j, e_{i,j})] x_j + \sum_{j \in K(i)} [\alpha s_{ij} + \beta p_{ij} + \phi(x_i, x_j)] x_j \\
&= \sum_{j \in N(i)} a_{ij} x_j + \sum_{j \in K(i)} a_{ij} x_j
\end{aligned}
$$

There is a normalization method so that each row of A is equal to 1, which leads to that when GT and DGTAT have the same attention to important long-range dependencies, the nodes of DGTAT have relatively higher attention to themselves as well as to their neighboring nodes than GT, focusing on important long-range dependencies while better preserving their own information and avoiding the phenomenon of over-smoothing.

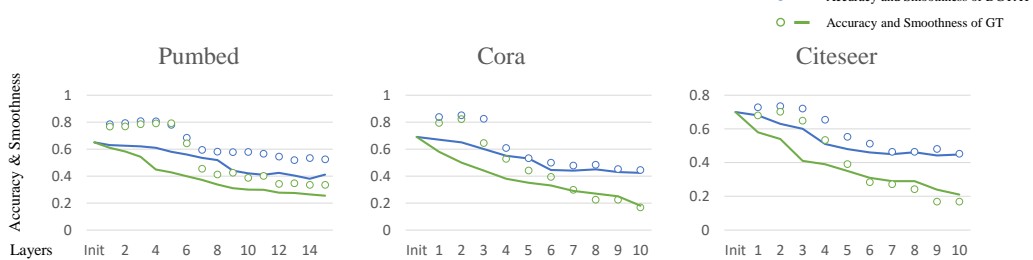

Figure 6: Smoothness metric value with different numbers of layers of DGTAT and GT

$$sm(\hat{G})_{DGTAT} = \sum_{i,j} \left\| \frac{\hat{x}_i}{||\hat{x}_i||} - \frac{\hat{x}_j}{||\hat{x}_j||} \right\|_{DGTAT}$$

$$= \sum_{i,j} ||\hat{x}_i - \hat{x}_j||_{DGTAT}$$

$$= \sum_{i,j} \sum_{x} ||\hat{x}_{ix} - \hat{x}_{jx}||_{DGTAT}$$

$$= \sum_{i,j} \sum_{x} \left\| \sum_{y \in N(x)} x_{iy} A_{N\,yx} + \sum_{y \in K(x)} x_{iy} A_{S\,yx} - \sum_{y \in N(x)} x_{jy} A_{N\,yx} - \sum_{y \in K(x)} x_{jy} A_{S\,yx} \right\|$$

$$= \sum_{i,j} \sum_{x} \left\| \sum_{y \in N(x)} (x_{iy} - x_{jy}) A_{N\,yx} + \sum_{y \in K(x)} (x_{iy} - x_{jy}) A_{S\,yx} \right\|$$

$$= \sum_{i,j} \sum_{x} \sum_{y \in N(x)} ||(x_{iy} - x_{jy})|| A_{N\,yx} + \sum_{i,j} \sum_{x} \sum_{y \in K(x)} ||(x_{iy} - x_{jy})|| A_{S\,yx}$$

$$\geq \sum_{i,j} \sum_{x} \sum_{y \in N(x) \cup K(x)} ||(x_{iy} - x_{jy})|| (A_{N\,yx} + A_{S\,yx})$$

$$\geq \sum_{i,j} \sum_{x} \sum_{y \in V} ||(x_{iy} - x_{jy})|| A_{yx}$$

$$= \sum_{i,j} \sum_{x} \left\| \sum_{y \in V} x_{iy} A_{yx} - \sum_{y \in V} x_{jy} A_{yx} \right\|$$

$$= \sum_{i,j} ||\hat{x}_i - \hat{x}_j||_{GT}$$

$$= sm(\hat{G})_{GT}$$

The inequality holds if the nodes of DGTAT and GT have the same attention to long-range dependencies in $K(i)$, the $sm$ metric of DGTAT is larger than that of GT, proving that our DGTAT captures the long-range dependency while being further away from the over-smoothing than GT.

## D  A FAST ALGORITHM TO COMPUTE THE JACCARD ENCODING

Here we provide an algorithm to initialize the Jaccard encoding. Intuitively, the relative positional importance of the nodes should decay with their shortest distance from the center node. We adopt a simple idea that computes the weight of the node j using a Gaussian decay, i.e., $P_{i,j}^{Jaccard} = exp(-\frac{distance(i,j)^2}{2h^2})$.

---

**Algorithm 1** Fast Jaccard encoding computing

---

**Input:** Normalized adjacency matrix $\tilde{A}$; Positional sensing range K
**Output:** Initialized positional encoding $P^{Jaccard}$
1: Initialization:$P^{Jaccard} \leftarrow 0, B = I$
2: **for** $k = 0$ to $K - 1$ **do**
3:     **for** $i, j$ in range($N$) **do**
4:         **if** $B_{i,j} > 0$ and $P_{i,j}^{Jaccard} = 0$ **then**
5:             $P_{i,j}^{Jaccard} = exp(-\frac{k^2}{2})$
6:         **end if**
7:     **end for**
8:     $B = B\tilde{A}$
9: **end for**
10: **return** $P^{Jaccard}$

---

## E  COMPLEXITY ANALYSIS OF DGTAT

**Time complexity**

The time complexity of DGTAT mainly depends on 3 modules. The complexity of the global attention module for structural and positional encodings is $O(n^2k)$ and we can use a linear kernel attention method to reduce it to $O(nk)$. The complexity of attribute attention and message passing is $O((|\mathcal{E}| + nk)d)$. The computation of RRWP and RRSE is $O(nk|\mathcal{E}|)$, where $n$ denotes the number of nodes, $d$ denotes the feature dimension, $|\mathcal{E}|$ denotes the number of edges and $k$ denotes the hyperparameter (the length of the PE/SE and the number of sampling nodes each layer).

**Space complexity**

The space complexity is based on the number of model parameters and the outputs of each layer. The first part is mainly on the structural and positional attention module $O(dk)$. The second part is on the edge updating methods $O((d + d_e)d')$. And there are some MLP or FFN layers cost $O(dk)$ or $O(dd')$. Thus, the total space complexity in one layer is $O(dk + (d + d_e)d')$, where $d'$ denotes the hidden dimension and $d_e$ denotes the dimension of edge feature.

**Complexity compared to baselines**

The source of Transformer's complexity is mainly the global computation of attention, which has a quadratic complexity in terms of the number of nodes. In fact, although we compute triple attention, our complexity is even lower than that of many GT-based baselines, mainly because we do not compute global attention for attribute encoding.

During the sampling phase, we only compute global attention for positional encoding and structural encoding, whereas standard GTs computes global attention using node attribute encoding. The length of the positional encodings and structural encodings (usually no more than 12) is much smaller than the length of the attribute encoding (the initial feature is larger than 1K and the hidden dimension is usually at least 64).

During the attribute information propagation and update phase of DGTAT, nodes only communicate with neighboring nodes, sampled nodes and virtual nodes, for which the complexity of the attribute encoding is linear.

Therefore, the maximum complexity of DGTAT is smaller than many GT baselines, as the table 10 shown. Moreover, some work has used linear transformer methods to reduce the complexity of their models theoretically to linear, e.g. GraphGPS uses Performer, but in fact, experiments have shown better performance using the standard transformer. In future work, we can also further reduce the complexity by optimizing with these linear GT approach.

Table 10: Complexity compared to baselines, where $n$ is the number of nodes and $d$ is the dimension of attribute encoding

|  | Complexity | Notes |
|---|---|---|
| GCN-II | $O(nd)$ | |
| GAT | $O(nd)$ | |
| GT | $O(n^2d)$ | |
| Graphormer | $O(n^2d)$ | |
| GraphGPS | $O(nd)$ or $O(n^2d)$ | Linear if using Performer, quadratic if using full Transformer |
| NAGphormer | $O(n(K+1)^2d)$ | K is the number of hops |
| DGTAT | $O(n^2K + nd)$ | K is the length of PE/SE and the number of sampled nodes each layer and $K << d$ |

# F  QUESTIONS AND RESPONSE

This section summarises our explanations and responses to the weaknesses and questions from reviewers and is used to facilitate the resolution of questions and further discussion with us. These additions and modifications have been updated to our paper.

**W1: The sampling strategy based on positional/structural attention is heuristic and may not optimally capture long-range dependencies.**

Our motivation for proposing the sampling strategy is to improve the long-range dependency capturing capability of the model. Through the global attention computation of positional and structural encoding, we sample nodes with similar positional receptive field and structural topology without distance constraints, and connect them for communication to capture long-range dependencies.

While the sampling strategy is heuristic, each training layer samples K optimal long-range-dependency nodes for communication. For different datasets, different number of layers and hyperparameters K are selected to ensure that every important long-range dependency is captured. We remain open to exploring potential refinements and extensions in future research.

Additionally, the introduction of virtual nodes improves the ability to capture long-range dependencies from attribute information. Therefore, DGTAT has a good ability to optimally capture long-range dependencies in all three aspects: attribute information, location information, and structural information.

From an experimental point of view, large-scale graphs require more of a model's ability to capture long-range dependencies, and thus a model's performance on large-scale graphs is an important metric for evaluating a model's ability to capture long-range dependencies. Our experimental results exemplify that DGTAT achieves significant accuracy improvement relative to SOTA baselines.

Additionally, we have added an ablation experiment for the sampling strategy, shown as 7. We test DGTAT without the sampling strategy as a comparison (other computations are consistent with standard DGTAT). The experimental results show a significant decrease in accuracy for all datasets tested, especially for large-scale graphs, further illustrating the ability of our sampling strategy to capture long-range dependencies.

**Q1: How is the sampling distribution optimized during training? Is there a learnable component?**

As shown in the paper in Figure 1, we have a crucial learnable parameter that dynamically weighs the importance of structural and positional attention to optimize our sampling strategy. It allows us to learn to measure whether long-range dependencies between different nodes in the dataset are more correlated with positional information or more correlated with structural information, so that our model is well adapted to different datasets.

Furthermore, the learnability of the structural encoding and positional encoding is also relevant for optimizing sampling. As these encodings are learned and updated at each layer, they influence attention computation, subsequently shaping the sampling results and ultimately contributing to an optimized sampling strategy.

**W2: Increased model complexity due to decoupled computations and triple attention.**

The source of Transformer's complexity is mainly the global computation of attention, which has a quadratic complexity in terms of the number of nodes. In fact, although we compute triple attention, our complexity is even lower than that of standard GT, mainly because we do not compute global attention for attribute encoding.

During the sampling phase, we only compute global attention for positional encoding and structural encoding, whereas standard GTs computes global attention using node attribute information. The length of the positional encodings and structural encodings (usually no more than 12) is much smaller than the length of the attribute encoding (the initial feature is larger than 1K and the hidden dimension is usually at least 64).

During the attribute information propagation and update phase of DGTAT, nodes only communicate with neighboring nodes, sampled nodes and virtual nodes, for which the complexity of the attribute encoding is linear.

Therefore, the maximum complexity of DGTAT is smaller than standard GT. In future work, we can further reduce the complexity by optimizing with the linear GT approach, such as using the methods of performer, etc.

**W3 & Q2: What is the empirical complexity of DGTAT compared to GT and MPNN baselines?**

Appendix E of the original paper contains a complexity analysis for DGTAT, and in order to evaluate the complexity of DGTAT more concretely and intuitively, we have added an ablation experiment as table 10 in Appendix E to compare the complexity of DGTAT with various GT-based and MPNN.

**W4: The contribution of the paper is marginal. The performance improvement is most likely due to the incorporation of existing techniques like MPNN+VN, LapPE, JaccardPE, SE.**

As you said, some of the strategies we used such as MPNN+VN, LapPE, RRWP, and RRSE do already exist. They were proposed by some previous works with good results.

However, the previous works did not delve deeper into exactly where each encoding contributes, e.g., acting on local structural information/global structural information/local position information/global position information, and how they can be incorporated.

Moreover, it turns out that simply stacking and incorporating these strategies and encodings does not improve performance further due to the coupling of information (We give an analysis in the Introduction, and in our revised paper we have added ablation experiments that support this claim, as described in the next question and table 9). So, it is important to find a way to incorporate these techniques to improve the performance of graph learners.

Our idea and work solve this problem, and are evaluated as interesting and solid by other reviewers. We analyzed the specific domains in which these techniques play a role. For the first time in the GT domain, we propose the idea of decoupling structure, position and attribute information, presenting a clean decoupled framework.

By populating our framework with already existing encodings that contain decoupled information, combined with our sampling strategy and triple-attention method, we effectively combine these encodings to make them work decoupledly and achieve SOTA performance. So, indeed, what we have done matters, and getting these technologies to incorporate well is where we excel.

**W5 & Q3: Why we need decoupling in graphs is not well discussed. There are only vague claims on page 14 stating that "compared to the GT with SE/PE, through the decoupling of PE, SE and AE, DGTAT can distinguish some graphs with more sensitive position, structure, and attribute information that coupled PS/SE cannot learn" (this is an assumption, not an explanation/analysis). Figure 5 also cannot support this claim well.**

We appreciate your insightful observation regarding the need for a more thorough discussion on the importance of decoupling. This is a noteworthy issue, in the Introduction we analyzed the need for decoupling, in Appendix B we illustrated the expressive power around DGTAT, and in Figure5 we illustrated the role of decoupling by showing some cases that only decoupled encodings can yield correct classification results for nodes.

As you say, this may not be enough to support our claim well. After submitting our original paper we have recognized the necessity of adding more well-designed ablation experiments in Appendix A to demonstrate the role of decoupling, and we have supported our claim through these experiments.

Exp2 and 3 further support the importance of the decoupling strategy; Exp4 illustrates the crucial role of our design framework and shows that the performance improvement is not due to a simple stacking of encodings. Specifically, as shown in Appendix A: More ablation experiments. Results are shown in table 7, 8, and 9.

**W6: Only node classification experiments are performed. In contrast, standard GTs are evaluated by graph classification.**

The node classification task is one of the most classical and mainstream tasks in the graph domain, and one of our motivations is to capture long-range dependencies, which requires the evaluation of medium-scale and large-scale graphs, which are generally employed in the node classification task. In addition, the node classification task can better reflect the generality of DGTAT for both heterophilic and homophilic graphs.

Most of the baseline papers of GT have been evaluated on the node classification task, and some work also evaluated only the node classification task (such as NodeFormer, NAGPHORMER, and AERO-GNN). Similarly, we only consider the graph classification task and focus on the node classification task.

It is important to note that this does not mean that DGTAT does not perform well in the graph classification task. To allay your concerns, we conducted a preliminary experiment on a graph classification dataset. By using virtual nodes with a pooling readout strategy, we extended DGTAT to graph classification tasks with SOTA results in the dataset, shown as follows, and more graph classification tasks will be our follow-up work.

Table 11: Result in benchmark ZINC. Shown is the mean ± s.d. of 10 runs with different random seeds. Here DGTAT uses a virtual node pooling readout strategy to extend to graph classification tasks.

|  | MAE ↓ |
| --- | --- |
| GCN | 0.367 ± 0.011 |
| GAT | 0.384 ± 0.007 |
| SAN | 0.139 ± 0.006 |
| GT | 0.598 ± 0.049 |
| Graphprmer | 0.122 ± 0.006 |
| K-Subgraph SAT | 0.094 ± 0.008 |
| GraphGPS | 0.070 ± 0.004 |
| DGTAT | 0.059 ± 0.004 |

**W7: The major concern is the experimental result of the proposed method. According to Table 3 and Table 4, the proposed method does not show significant improvement over baseline methods. Most of the numbers are quite close to the previous SOTA results. It is unclear to see from the results that the proposed techniques in this paper works.**

DGTAT shows a steady improvement over the SOTA baseline on the datasets listed in our paper, both in terms of maximum and average accuracy (Accuracy improvements ranged from 0.14% to 1.98%, with an average improvement of 1%). Compared to the GT-based SOTA baseline, our improvement is sufficiently significant that it is greater than or equal to the magnitude of the improvement in SOTA on these datasets by previously published GT-related works (such as GRIT, GraphGPS, etc), especially on large-scale graphs that require long-range dependency (compare with the SOTA baseline: NAGphormer).

The clean decoupled framework we propose and the results obtained from our ablation experiments, in which each decoupled component contributes, are more noteworthy. This can inspire a deeper exploration of the decoupling idea, searching for a coding that is more in line with the design idea, as well as exploring the theoretical relationship between the graph structure information and the location information.

In response to "It is unclear to see from the results that the proposed techniques in this paper works.", we have included more ablation experiments in Appendix A to demonstrate that it is our decoupled framework and our proposed method that allows DGTAT to achieve significantly higher performance than GT baselines, as shown in our response to the next question.

**W8: There is no experimental result showing that how the proposed components in this paper performs. It is necessary to do ablation study to see how the performance changes with/without a particular module.**

As mentioned by one reviewer as a strength, there are already well-designed ablation experiments in our original submission (Section 4.3, Figure3 and Table5), which demonstrate that each of our decoupled modules and strategies such as positional encoding, structural encoding, virtual nodes, and learnability of these encodings can contribute an improvement in the overall DGTAT.

In our already refined work, we have designed more detailed ablation experiments: Exp1 illustrates the contribution of our sampling strategy; Exp2 and 3 further support the importance of the decoupling strategy; Exp4 illustrates the crucial role of our design framework and shows that the performance improvement is not due to a simple stacking of encodings. Specifically, as shown in Appendix A: More ablation experiments. Results are shown in table 7, 8, and 9.

**W9: The presentation of this paper could be improved. The authors introduces a lot of notations, and sometimes it is hard to find the actual meaning of a particular notation. Figure 1 and figure 2 is also confusing and hard to understand.**

We sincerely appreciate your constructive feedback. We have improved the writing in the revised paper. And there was a clerical error in the original Figure 1's annotation, which we have corrected.

In addition, we have improved the annotations for Figures 1 and 2 by explaining their meaning in more detail, and we have reviewed and revised all our symbols, formulae and annotations to make them clearer and easier to understand.

