# OpenReview forum: "DGTAT: DECOUPLED GRAPH TRIPLE ATTENTION NETWORKS"
_ICLR.cc/2024/Conference — ICLR 2024 Conference Withdrawn Submission_

### Official Review · Reviewer_xSa6 · 2023-10-31

**Soundness:** 3 good
**Presentation:** 2 fair
**Contribution:** 2 fair
**Rating:** 5
**Confidence:** 4

**Summary:**

This paper analyzes the factors influencing the performance of graph learning models. The authors proposes a model named DGTAT, which is based on MPNN and a sampling strategy. The proposed method decouples local and global interactions, separates learnable positional, attribute, and structural encodings, and computes triple attention. This design allows DGTAT to capture long-range dependencies akin to Transformers while preserving the inductive bias of the graph topology.

**Strengths:**

1. The idea of decoupling local and global interactions, as well as separating learnable positional, attribute, and structural encodings, is interesting;

2. The proposed method combines several techniques, i.e. laplacian, random walk, positional encoding, GNN, which is solid;

**Weaknesses:**

1. The major concern is the experimental result of the proposed method. According to Table 3 and Table 4, the proposed method does not show significant improvement over baseline methods. Most of the numbers are quite close to the previous SOTA results. It is unclear to see from the results that the proposed techniques in this paper works.

2. There is no experimental result showing that how the proposed components in this paper performs. It is necessary to do ablation study to see how the performance changes with/without a particular module.

3. The presentation of this paper could be improved. The authors introduces a lot of notations, and sometimes it is hard to find the actual meaning of a particular notation. Figure 1 and figure 2 is also confusing and hard to understand.

**Questions:**

see weaknesses above

---

> ### Author Response · Authors · 2023-11-15
> **Reply (1/3)**
>
> We appreciate your effort very much in commenting on our paper and endorsement of our idea and methods. We will explain the weaknesses that you put forward and answer your questions.
> >###  W1. The major concern is the experimental result of the proposed method. According to Table 3 and Table 4, the proposed method does not show significant improvement over baseline methods. Most of the numbers are quite close to the previous SOTA results. It is unclear to see from the results that the proposed techniques in this paper works.
>
> DGTAT shows a steady improvement over the SOTA baseline on the datasets listed in our paper, both in terms of maximum and average accuracy (Accuracy improvements ranged from 0.14\% to 1.98\%, with an average improvement of 1\%). Compared to the GT-based SOTA baseline, our improvement is sufficiently significant that it is greater than or equal to the magnitude of the improvement in SOTA on these datasets by previously published GT-related works (such as GRIT, GraphGPS, etc), especially on large-scale graphs that require long-range dependency (compare with the SOTA baseline: NAGphormer).
>
> The clean decoupled framework we propose and the results obtained from our ablation experiments, in which each decoupled component contributes, are more noteworthy. This can inspire a deeper exploration of the decoupling idea, searching for a coding that is more in line with the design idea, as well as exploring the theoretical relationship between the graph structure information and the location information.
>
> In response to *"It is unclear to see from the results that the proposed techniques in this paper works."*, we have included more ablation experiments in Appendix A to demonstrate that it is our decoupled framework and our proposed method that allows DGTAT to achieve significantly higher performance than GT baselines, as shown in our response to the next question.

---

> > ### Author Response · Authors · 2023-11-15
> > **Reply (2/3)**
> >
> > >### W2. There is no experimental result showing that how the proposed components in this paper performs. It is necessary to do ablation study to see how the performance changes with/without a particular module.
> >
> > As mentioned by one reviewer as a strength, there are already well-designed ablation experiments in our original submission (Section 4.3, Figure 3 and Table 5), which demonstrate that each of our decoupled modules and strategies such as positional encoding, structural encoding, virtual nodes, and learnability of these encodings can contribute an improvement in the overall DGTAT.
> >
> > In our already refined work, we have designed more detailed ablation experiments: Exp1 illustrates the contribution of our sampling strategy; Exp2 and 3 further support the importance of the decoupling strategy; Exp4 illustrates the crucial role of our design framework and shows that the performance improvement is not due to a simple stacking of encodings. Specifically, shown as follows:
> >
> > Exp1: **Remove DGTAT’s sampling strategy** (other computations are consistent with standard DGTAT). As shown in the table, the experimental results show a significant decrease in accuracy for all datasets tested, especially for large-scale graphs, further illustrating the ability of the sampling strategy to capture long-range dependencies.
> > |                        | Pubmed  | Citeseer | Cora  | Texas  | Cornell   | Actor | AMiner-CS | Amazon2M |
> > |------------------------|---------|----------|-------|--------|-----------|-------|-----------|----------|
> > | DGTAT without sampling | 79.88   | 73.1     | 83.97 | 79.15  | 78.24     | 35.77 | 54.97     | 76.51    |
> > | DGTAT                  | 80.71   | 73.4     | 85.12 | 84.39  | 82.14     | 36.43 | 56.38     | 78.49    |
> > | Gain                   | +0.83   | +0.3     | +1.15 | +5.24  | +3.9      | +0.66 | +1.41     | +1.98    |
> >
> > Exp2: **Couple the structural encoding and positional encoding in the DGTAT framework.** Unlike standard DGTAT, this experiment splices or sums the normalized structural encoding and positional encoding into a coupled encoding. Then we use the coupled encoding to compute global attention, and then sample long-range-dependency based on the results. Experimental results show significantly lower performance than standard DGTAT, illustrating the positive contribution of the decoupling of positional encoding and structural encoding.
> >
> > EXP3: **Couple the structural encoding and positional encoding into the attribute features in the DGTAT framework.** We use this structure, position, and attribute coupled encoding to compute global attention, and then sample based on the results. A more pronounced decrease in performance occurs, illustrating the positive contribution of the decoupling of attribute encoding and other encodings.
> >
> > The results of Exp2 and 3 are shown as follows:
> > |                             | Pubmed          | Citeseer      | Cora            |
> > |-----------------------------|-----------------|---------------|-----------------|
> > | DGTAT                       | 80.71           | 73.4          | 85.12           |
> > | DGTAT with coupled SE+PE    | 79.94 ($-$0.75) | 72.4 ($-$1.0) | 82.44 ($-$2.68) |
> > | DGTAT with coupled SE+PE+AE | 78.96 ($-$1.75) | 71.8 ($-$1.6) | 82.76 ($-$2.36) |
> >
> > EXP4: In some baseline models of MPNN and GT (specifically, GATv2 and GPS), we **directly incorporate the different encodings** used in our work. With the addition and coupling of encodings, the performance does not improve or improves only weakly. Even the direct incorporation of these coupled encodings instead leads to a severe decrease in accuracy. This experimental result illustrates that leaving the decoupling framework of DGTAT, simple stacking of encodings cannot improve GT performance, further supporting that our idea of decoupling makes contribution and our framework is solid.
> >
> > The results of Exp4 are shown as follows:
> > |                          | Pubmed          | Citeseer      | Cora            |
> > |--------------------------|-----------------|---------------|-----------------|
> > | GATv2                    | 79.22           | 72.1          | 83.95           |
> > | GATv2+LapPE              | 79.27 ($+$0.05) | 72.3 ($+$0.2) | 84.21 ($+$0.26) |
> > | GATv2+LapPE+RRSE         | 78.96 ($-$0.26) | 71.9 ($-$0.2) | 83.72 ($-$0.23) |
> > | GATv2+LapPE+RRWP+RRSE    | 77.61 ($-$1.61) | 69.8 ($-$2.3) | 80.53 ($-$3.42) |
> > | GraphGPS (without PE/SE) | 79.58           | 72.1          | 82.26           |
> > | GraphGPS+LapPE           | 79.94 ($+$0.36) | 72.4 ($+$0.3) | 82.44 ($+$0.18) |
> > | GraphGPS+LapPE+RRSE      | 80.53 ($+$0.95) | 72.9 ($+$0.8) | 83.29 ($+$1.03) |
> > | GraphGPS+LapPE+RRWP+RRSE | 79.09 ($-$0.49) | 71.9 ($-$0.5) | 82.57 ($+$0.31) |
> >
> > We have added the content of these ablation experiments to our Appendix A: More Experimental Details. As well, we have summarized them Appendix F of the revised paper to make them more accessible to you.

---

> > > ### Author Response · Authors · 2023-11-15
> > > **Reply (3/3)**
> > >
> > > >### W3. The presentation of this paper could be improved. The authors introduces a lot of notations, and sometimes it is hard to find the actual meaning of a particular notation. Figure 1 and figure 2 is also confusing and hard to understand.
> > >
> > > We sincerely appreciate your constructive feedback. We have improved the writing in the revised paper. And there was a clerical error in the original Figure 1’s annotation, which we have corrected.
> > >
> > > In addition, we have improved the annotations for Figures 1 and 2 by explaining their meaning in more detail, and we have reviewed and revised all our symbols, formulae and annotations to make them clearer and easier to understand.
> > >
> > >
> > > Many thanks again for your devotion and comments!  All the updates and modifications are now added to our revised paper. And we summed up all the comments on this paper from every reviewer including yours and our responses to them in Appendix F in our revised paper.
> > >
> > > We appreciate it very much if you could read our updates. We always remain available to answer any other questions and suggestions on our paper and improve it. We are looking forward to your reply and new reviewing grade.

---

> ### Author Response · Authors · 2023-11-21
> **A kindly Reminder Message: 2days left in discussion phase**
>
> Dear Reviewer xSa6:
>
> I hope this email finds you well. As the deadline for revision and discussion is approaching, we are sincerely looking forward to your feedback.
>
> We understand you may have a busy schedule, but we believe that we have addressed all your concerns and refined our paper to ensure that our work meets the standards.
>
> If you still have further concerns or feel unclear after reading our responses, please do not hesitate to reach out to us. We would be happy to have a further discussion with you. If you are satisfied with our responses so far, we sincerely hope you could consider your score.
>
> Thank you in advance!
>
> Best wishes, Authors

---

> > ### Author Response · Authors · 2023-11-23
> > **Looking forward to a discussion before the deadline**
> >
> > Dear reviewer xSa6:
> >
> > I hope you’re having a great day. Since there are only less than **10 hours** to the deadline for the discussion phase, we are really looking forward to having a discussion with you. We sincerely hope to get your further feedback. Would you mind checking our response and letting us know if you have further questions? If you are satisfied with our responses so far, we sincerely hope you could consider your score, which matters to us. Thank you in advance for your time!
> >
> > Best regards, Authors of #878

---

### Official Review · Reviewer_Zxgv · 2023-10-31

**Soundness:** 3 good
**Presentation:** 2 fair
**Contribution:** 2 fair
**Rating:** 5
**Confidence:** 3

**Summary:**

This paper proposes to decouple (position, structure, attribute) and (global interaction local interaction) in graph transformers with a novel model DGTAT. DGTAT is consistently effective on both Homophilic and heterophilic graphs at different scales.

**Strengths:**

S1 The proposed method is consistently effective on both Homophilic and heterophilic graphs at different scales.

S2 The proposed method avoids over-smoothing and has better expressivity.

**Weaknesses:**

W1 The contribution of the paper is marginal. The performance improvement is most likely due to the incorporation of existing techniques like MPNN+VN, LapPE, JaccardPE, SE.

W2 Why we need decoupling in graphs is not well discussed. There are only vague claims on page 14 stating that "compared to the GT with SE/PE, through the decoupling of PE, SE and AE, DGTAT can distinguish some graphs with more sensitive position, structure, and attribute information that coupled PS/SE cannot learn" (this is an assumption, not an explanation/analysis). Figure 5 also cannot support this claim well.

W3. Only node classification experiments are performed. In contrast, standard GTs are evaluated by graph classification.

**Questions:**

Explain W2 please.

---

> ### Author Response · Authors · 2023-11-15
> **Reply (1/3)**
>
> We appreciate your effort very much in commenting on our paper. Firstly we would like to clarify certain misunderstandings in W1. Then we acknowledge your concerns in W2 and have added sufficient ablation experiments in our refined work to resolve the questions of W2. At last we conducted a preliminary experiment to alley your concern in W3.
>
> >### W1. The contribution of the paper is marginal. The performance improvement is most likely due to the incorporation of existing techniques like MPNN+VN, LapPE, JaccardPE, SE.
>
> As you said, some of the strategies we used such as MPNN+VN, LapPE, RRWP, and RRSE do already exist. They were proposed by some previous works with good results.
>
> However, the previous works did not delve deeper into exactly where each encoding contributes, e.g., acting on local structural information/global structural information/local position information/global position information, and how they can be incorporated.
>
> Moreover, it turns out that simply stacking and incorporating these strategies and encodings does not improve performance further due to the coupling of information (We give an analysis in the Introduction, and in our revised paper we have added ablation experiments that support this claim, as described in the response to W2). So, it is important to find a way to incorporate these techniques to improve the performance of graph learners.
>
> **Our idea and work solve this problem, and are evaluated as interesting and solid by other reviewers.** We analyzed the specific domains in which these techniques play a role. **For the first time in the GT domain, we propose the idea of decoupling structure, position and attribute information, presenting a clean decoupled framework.**
>
> By populating our framework with already existing encodings that contain decoupled information, combined with our sampling strategy and triple-attention method, we effectively combine these encodings to make them work decoupledly and achieve SOTA performance. So, indeed, what we have done matters, and getting these technologies to incorporate well is where we excel.

---

> > ### Author Response · Authors · 2023-11-15
> > **Reply (2/3)**
> >
> > >### W2. Why we need decoupling in graphs is not well discussed. There are only vague claims on page 14 stating that "compared to the GT with SE/PE, through the decoupling of PE, SE and AE, DGTAT can distinguish some graphs with more sensitive position, structure, and attribute information that coupled PS/SE cannot learn" (this is an assumption, not an explanation/analysis). Figure 5 also cannot support this claim well.
> >
> > We appreciate your insightful observation regarding the need for a more thorough discussion on the importance of decoupling. This is a noteworthy issue, in the Introduction we analyzed the need for decoupling, in the Appendix B we illustrated the expressive power around DGTAT, and in Figure5 we illustrated the role of decoupling by showing some cases that only decoupled encodings can yield correct classification results for nodes.
> >
> > As you say, this may not be enough to support our claim well. After submitting our original paper we have recognized the necessity of adding more well-designed ablation experiments to demonstrate the role of decoupling, and we have supported our claim through these experiments.
> >
> > Exp2 and 3 further support the importance of the decoupling strategy; Exp4 illustrates the crucial role of our design framework and shows that the performance improvement is not due to a simple stacking of encodings. Specifically, as follows:
> >
> > Exp2: **Couple the structural encoding and positional encoding in the DGTAT framework.** Unlike standard DGTAT, this experiment splices or sums the normalized structural encoding and positional encoding into a coupled encoding. Then we use the coupled encoding to compute global attention, and then sample long-range-dependency based on the results. Experimental results show significantly lower performance than standard DGTAT, illustrating the positive contribution of the decoupling of positional encoding and structural encoding.
> >
> > EXP3: **Couple the structural encoding and positional encoding into the attribute features in the DGTAT framework.** We use this structure, position, and attribute coupled encoding to compute global attention, and then sample based on the results. A more pronounced decrease in performance occurs, illustrating the positive contribution of the decoupling of attribute encoding and other encodings.
> >
> > EXP4: In some baseline models of MPNN and GT (specifically, GATv2 and GPS), we **directly incorporate the different encodings** used in our work. With the addition and coupling of encodings, the performance does not improve or improves only weakly. Even the direct incorporation of these coupled encodings instead leads to a severe decrease in accuracy. This experimental result illustrates that leaving the decoupling framework of DGTAT, simple stacking of encodings cannot improve GT performance, further supporting that our idea of decoupling makes contribution and our framework is solid.
> >
> > The results of Exp2 and 3 are shown as follows:
> > |                             | Pubmed          | Citeseer      | Cora            |
> > |-----------------------------|-----------------|---------------|-----------------|
> > | DGTAT                       | 80.71           | 73.4          | 85.12           |
> > | DGTAT with coupled SE+PE    | 79.94 ($-$0.75) | 72.4 ($-$1.0) | 82.44 ($-$2.68) |
> > | DGTAT with coupled SE+PE+AE | 78.96 ($-$1.75) | 71.8 ($-$1.6) | 82.76 ($-$2.36) |
> >
> > The results of Exp4 are shown as follows:
> > |                          | Pubmed          | Citeseer      | Cora            |
> > |--------------------------|-----------------|---------------|-----------------|
> > | GATv2                    | 79.22           | 72.1          | 83.95           |
> > | GATv2+LapPE              | 79.27 ($+$0.05) | 72.3 ($+$0.2) | 84.21 ($+$0.26) |
> > | GATv2+LapPE+RRSE         | 78.96 ($-$0.26) | 71.9 ($-$0.2) | 83.72 ($-$0.23) |
> > | GATv2+LapPE+RRWP+RRSE    | 77.61 ($-$1.61) | 69.8 ($-$2.3) | 80.53 ($-$3.42) |
> > | GraphGPS (without PE/SE) | 79.58           | 72.1          | 82.26           |
> > | GraphGPS+LapPE           | 79.94 ($+$0.36) | 72.4 ($+$0.3) | 82.44 ($+$0.18) |
> > | GraphGPS+LapPE+RRSE      | 80.53 ($+$0.95) | 72.9 ($+$0.8) | 83.29 ($+$1.03) |
> > | GraphGPS+LapPE+RRWP+RRSE | 79.09 ($-$0.49) | 71.9 ($-$0.5) | 82.57 ($+$0.31) |
> >
> > We have added the content of these ablation experiments to our Appendix A: More Experimental Details. As well, we have summarized them Appendix F of the revised paper to make them more accessible to you.

---

> ### Author Response · Authors · 2023-11-15
> **Reply (3/3)**
>
> >### W3. Only node classification experiments are performed. In contrast, standard GTs are evaluated by graph classification.
>
> The node classification task is one of the most classical and mainstream tasks in the graph domain, and one of our motivations is to capture long-range dependencies, which requires the evaluation of medium-scale and large-scale graphs, which are generally employed in the node classification task. In addition, the node classification task can better reflect the generality of DGTAT for both heterophilic and homophilic graphs.
>
> Most of the baseline papers of GT have been evaluated on the node classification task, and some work also evaluated only the node classification task (such as NodeFormer, NAGPHORMER, and AERO-GNN). Similarly, we only consider the graph classification task and focus on the node classification task.
>
> It is important to note that this does not mean that DGTAT does not perform well in the graph classification task. To allay your concerns, we conducted a preliminary experiment on a graph classification dataset. By using virtual nodes with a pooling readout strategy, we extended DGTAT to graph classification tasks with SOTA results in the ZINC dataset, shown as follows, and more graph classification tasks will be our follow-up work.
> |                | MAE ↓         |
> |----------------|---------------|
> | GCN            | 0.367 ± 0.011 |
> | GAT            | 0.384 ± 0.007 |
> | SAN            | 0.139 ± 0.006 |
> | GT             | 0.598 ± 0.049 |
> | Graphprmer     | 0.122 ± 0.006 |
> | K-Subgraph SAT | 0.094 ± 0.008 |
> | GraphGPS       | 0.070 ± 0.004 |
> | DGTAT          | 0.059 ± 0.004 |
>
> Many thanks again for your devotion and comments!  All the updates and modifications are now added to our revised paper. And we summed up all the comments on this paper from every reviewer including yours and our responses to them in Appendix F in our revised paper.
>
> We appreciate it very much if you could read our rebuttals and updates. We always remain available to answer any other questions and suggestions on our paper and improve it. We are looking forward to your reply and new reviewing grade, which matters to us.

---

> > ### Comment · Reviewer_Zxgv · 2023-11-17
> > **Response**
> >
> > I sincerely appreciate the comprehensive response and additional results provided.
> > The authors have successfully convinced me these two aspects: (1) the essentiality of the framework's design and (2) the efficacy of DGTAT in graph classification tasks. Based on these insights, I have revised my evaluation accordingly.

---

> > > ### Author Response · Authors · 2023-11-17
> > > **Thanks for your feedback**
> > >
> > > Thank you for your swift feedback. If you have any further questions or suggestions, please do not hesitate to reach out to us. We would be happy to discuss with you.

---

### Official Review · Reviewer_2B2X · 2023-11-01

**Soundness:** 3 good
**Presentation:** 3 good
**Contribution:** 3 good
**Rating:** 5
**Confidence:** 4

**Summary:**

This paper proposes a novel graph neural network architecture called DGTAT that decouples the propagation of positional, structural, and attribute information to improve model expressiveness and interpretability. The key ideas are: 1) Use dedicated encodings to represent positional and structural information separately from attributes; 2) Compute triple attention based on positional, structural, and attribute information; 3) Sample relevant nodes based on positional/structural attention for capturing long-range dependencies. Experiments show SOTA results on node classification tasks.

**Strengths:**

- Clearly motivates the need for decoupling different types of information in GNNs, both theoretically and empirically.
- Proposes a clean design framework to achieve decoupling of positional, structural, and attribute information.
- Achieves SOTA results across multiple datasets, especially on heterophilic graphs.
- Ablation study illustrates the contribution of each decoupled component.

**Weaknesses:**

- The sampling strategy based on positional/structural attention is heuristic and may not optimally capture long-range dependencies.
- Increased model complexity due to decoupled computations and triple attention.
- Lacks analysis of computational efficiency compared to baselines.

**Questions:**

- How is the sampling distribution optimized during training? Is there a learnable component?
- What is the empirical complexity of DGTAT compared to GT and MPNN baselines?

---

> ### Author Response · Authors · 2023-11-15
> **Reply (1/2)**
>
> We appreciate your effort very much in commenting on our paper and endorsement of our innovations. We will explain the weaknesses that you put forward and answer your questions.
> >###  W1. The sampling strategy based on positional/structural attention is heuristic and may not optimally capture long-range dependencies.
>
> Our motivation for proposing the sampling strategy is to improve the long-range dependency capturing capability of the model. Through the global attention computation of positional and structural encoding, we sample nodes with similar positional receptive field and structural topology without distance constraints, and connect them for communication to capture long-range dependencies.
>
> While the sampling strategy is heuristic, each training layer samples K optimal long-range-dependency nodes for communication. For different datasets, different number of layers and hyperparameters K are selected to ensure that every important long-range dependency is captured. We remain open to exploring potential refinements and extensions in future research.
>
> Additionally, the introduction of virtual nodes improves the ability to capture long-range dependencies from attribute information. Therefore, DGTAT has a good ability to optimally capture long-range dependencies in all three aspects: attribute information, location information, and structural information.
>
> From an experimental point of view, large-scale graphs require more of a model's ability to capture long-range dependencies, and thus a model's performance on large-scale graphs is an important metric for evaluating a model's ability to capture long-range dependencies. Our experimental results exemplify that DGTAT achieves significant accuracy improvement relative to SOTA baselines.
>
> Additionally, we have added an ablation experiment for the sampling strategy. We test DGTAT without the sampling strategy as a comparison (other computations are consistent with standard DGTAT). The experimental results as follow show a significant decrease in accuracy for all datasets tested, especially for large-scale graphs, further illustrating the ability of our sampling strategy to capture long-range dependencies.
>
> |                        | Pubmed  | Citeseer | Cora  | Texas  | Cornell   | Actor | AMiner-CS | Amazon2M |
> |------------------------|---------|----------|-------|--------|-----------|-------|-----------|----------|
> | DGTAT without sampling | 79.88   | 73.1     | 83.97 | 79.15  | 78.24     | 35.77 | 54.97     | 76.51    |
> | DGTAT                  | 80.71   | 73.4     | 85.12 | 84.39  | 82.14     | 36.43 | 56.38     | 78.49    |
> | Gain                   | +0.83   | +0.3     | +1.15 | +5.24  | +3.9      | +0.66 | +1.41     | +1.98    |
>
> >### Q1. How is the sampling distribution optimized during training? Is there a learnable component?
>
> As shown in the paper in Figure 1, we have a crucial learnable parameter that dynamically weighs the importance of structural and positional attention to optimize our sampling strategy. It allows us to learn to measure whether long-range dependencies between different nodes in the dataset are more correlated with positional information or more correlated with structural information, so that our model is well adapted to different datasets.
>
> Furthermore, the learnability of the structural encoding and positional encoding is also relevant for optimizing sampling. As these encodings are learned and updated at each layer, they influence attention computation, subsequently shaping the sampling results and ultimately contributing to an optimized sampling strategy.
>
> >### W2. Increased model complexity due to decoupled computations and triple attention.
>
> The source of Transformer's complexity is mainly the global computation of attention, which has a quadratic complexity in terms of the number of nodes. In fact, although we compute triple attention, our complexity is even lower than that of standard GT, mainly because we do not compute global attention for attribute encoding.
>
> During the sampling phase, we only compute global attention for positional encoding and structural encoding, whereas standard GTs computes global attention using node attribute information. The length of the positional encodings and structural encodings (usually no more than 12) is much smaller than the length of the attribute encoding (the initial feature is larger than 1K and the hidden dimension is usually at least 64).
>
> During the attribute information propagation and update phase of DGTAT, nodes only communicate with neighboring nodes, sampled nodes and virtual nodes, for which the complexity of the attribute encoding is linear.
>
> Therefore, the maximum complexity of DGTAT is smaller than standard GT. In future work, we can further reduce the complexity by optimizing with the linear GT method, such as using Performer, etc.

---

> ### Author Response · Authors · 2023-11-15
> **Reply (2/2)**
>
> >### W3. Lacks analysis of computational efficiency compared to baselines & Q2. What is the empirical complexity of DGTAT compared to GT and MPNN baselines?
>
> Appendix E  of the original paper contains a complexity analysis for DGTAT as follows：
>
> ***Time complexity***
>
> *The time complexity of DGTAT mainly depends on 3 modules. The complexity of the global attention module for structural and positional encodings is $O(n^{2}k)$ and we can use a linear kernel attention method to reduce it to $O(nk)$. The  complexity of attribute attention and message passing is $O((|\varepsilon|+nk)d)$. The computation of RRWP and RRSE is $O(nk|\varepsilon|)$, where $n$ denotes the number of nodes, $d$ denotes the feature dimension, $|\varepsilon|$ denotes the number of edges and $k$ denotes the hyperparameter (the length of the PE/SE and the number of sampling nodes each layer).*
>
> ***Space complexity***
>
> *The space complexity is based on the number of model parameters and the outputs of each layer. The first part is mainly on the structural and positional attention module $O(dk)$. The second part is on the edge updating methods $O((d+d_{e})d^{\prime})$. And there are some MLP or FFN layers cost $O(dk)$ or $O(dd^{\prime})$. Thus, the total space complexity in one layer is $O(dk + (d+d_{e})d^{\prime})$, where $d^{\prime}$ denotes the hidden dimension and $d_{e}$ denotes the dimension of edge feature.*
>
> In order to evaluate the complexity of DGTAT more concretely and intuitively, we have added an table in the appendix to compare the complexity of DGTAT with various GT-based and MPNN, as shown below：
>
> |            | Complexity         | Notes                                                                          |
> |------------|--------------------|--------------------------------------------------------------------------------|
> | GCN-II     | $O(nd)$              |                                                                                |
> | GAT        | $O(nd)$              |                                                                                |
> | GT         | $O(n^{2}d)$          |                                                                                |
> | Graphormer | $O(n^{2}d)$          |                                                                                |
> | GraphGPS   | $O(nd) or O(n^{2}d)$ | Linear if using Performer, quadratic if using full Transformer                 |
> | NAGphormer | $O(n(K+1)^{2}d)$     | K is the number of hops                                                        |
> | DGTAT      | $O(n^{2}K+nd)$       | K is the length of PE/SE and the number of sampled nodes each layer and $K<<d$ |
>
>
> Many thanks again for your devotion and comments! More details are included on the Appendix F of our revised paper. All the updates and modifications are now added to our revised paper. We summed up all the comments on this paper from every reviewer including yours and our responses to them on the Appendix F. Other than the modifications we made on the basis of your questions, we carried out new ablation experiments that gave us interesting results.
>
> We appreciate it very much if you could read our updates. We always remain available to answer any other questions and suggestions on our paper and improve it. We are looking forward to your reply and new reviewing grade.

---

> ### Author Response · Authors · 2023-11-21
> **A kindly Reminder Message: 2days left in discussion phase**
>
> Dear Reviewer 2B2X:
>
> Thanks again for your great effort in reviewing our paper! Since the deadline for revision and discussion is approaching, we sincerely look forward to your feedback.
>
> We understand you may have a busy schedule, but we believe that we have addressed all your concerns and refined our paper to ensure that our work meets higher standards.
>
> If you still have further concerns or feel unclear after reading our responses, please kindly let us know and we are willing to have a further discussion with you about all technical details and concerns. If you are satisfied with our responses so far, we sincerely hope you could consider your score.
>
> Thanks very much!
>
> Best regards, Authors

---

> > ### Author Response · Authors · 2023-11-23
> > **Looking forward to a discussion before the deadline**
> >
> > Dear reviewer 2B2X:
> >
> > I hope you’re having a great day. Since there are only less than **10 hours** to the deadline for the discussion phase, we are really looking forward to having a discussion with you. We sincerely hope to get your further feedback.
> >
> > Would you mind checking our response and letting us know if you have further questions? And we sincerely want to share with you the interesting results we have achieved in our new ablation experiments, as shown in reply (2/3) of our response to reviewer xSa6. If you are satisfied with our responses so far, we sincerely hope you could consider your score, which matters to us. Thank you in advance for your time!
> >
> > Best regards, Authors of #878

---

### Author Response · Authors · 2023-11-20
**Summary of the revision**

Dear Reviewers:

Hope everything is fine with you! We are sincerely grateful for your careful and meticulous comments, which prompted us to improve the manuscript. To make it easier for you to check and understand, we summarize the changes as follows:

* We have added an analysis of the complexity advantage of DGTAT. In Appendix E, we describe the complexity advantage of DGTAT comparing to standard GTs, and add a detailed table comparing the complexity of DGTAT with MPNN and GT baselines.

* In response to Reviewer Zxgv's concern about the contribution of DGTAT and the need for decoupling, we explain the necessity of the framework design and the importance of decoupling and our sampling techniques. Our additional ablation experiments in Appendix A support our claims.

* In Appendix A, we have designed more detailed ablation experiments: Exp1 illustrates the contribution of our sampling strategy; Exp2 and 3 further support the importance of the decoupling strategy; Exp4 illustrates the crucial role of our design framework and shows that the performance improvement is not due to a simple stacking of encodings.

* To allay the concerns of DGTAT’s efficacy in graph classification tasks, we conducted a preliminary experiment on a graph classification dataset. By using virtual nodes with a pooling readout strategy, we extended DGTAT to graph classification tasks with SOTA results in the ZINC dataset, shown in Table 11.

* We followed Reviewer xSa6's advice in presentation of the paper. We have checked the writing in the revised paper and improved the annotations for Figures 1 and 2 by explaining their meaning in more detail, and we have thoroughly checked our use of symbols, formulae and annotations to make them clearer and readable.

In addition to the above changes in the paper, the weaknesses and issues raised by each reviewer have been explained in detail below the response and summarized in Appendix F to make it easier for everyone to check.

Best Wish, Authors

---

> ### Author Response · Authors · 2023-11-23
> **Statements before deadline: a brief summary of our rebuttals**
>
> Dear all:
>
> We appreciate all the reviewers for their comments and valuable suggestions for us to improve our work. The reviewers' comments were gentle and positive to our ideas, methods, and experimental results. We are encouraged to see the recognition of our work such as ***“clearly motivated; clean framework; better expressivity; interesting idea; soild method;”***
>
> At the rebuttal phase, we believe that our responses could address all the concerns raised by reviewers, and we have refined our paper to ensure that our work meets higher standards. Reviewer Zxgv has acknowledged our rebuttal and improved our score.
>
> However, as most of the reviewers did not respond to us, we summarize the reviewers' concerns about the paper here and briefly outlined our responses so that everyone can easily see our explanations.
>
> The main concerns of reviewer 2B2X focused on the heuristics and optimization of the sampling strategy, and the complexity of DGTAT.
>
> * For the sampling strategy, we provide an explanation on the learnability of the sampling strategy and its adaptability on different datasets, explaining the reason why the heuristic sampling strategy captures the long-range dependence well. And we add an additional ablation experiment to illustrate the effectiveness of our sampling strategy.
>
> * For the complexity of DGTAT, despite it has multiple modules, the design framework is clean, intuitive, and well-explained. In terms of time complexity and space complexity, we illustrate that the experimental complexity outperforms the standard GT, and add a comparison with MPNN and GT baselines.
>
>  Reviewer Zxgv questioned the contribution of the work and the necessity of decoupling, and had concerns about the performance of DGTAT on graph classification tasks.
>
> * We demonstrate the importance of our decoupling strategy and the excellent contribution of our proposed design framework. We also performed preliminary experiments on the graph classification task. Our explanations have successfully convinced the reviewer. The reviewer acknowledged our response, recognized the importance of our decoupling idea and design framework.
>
> Reviewer xSa6's main concern was that the work does not show significant improvement over baseline methods, which we explained that our rise beyond SOTA was significant enough relative to previous work, and more importantly that we demonstrated that it is our idea, framework and methods that make the contribution.
>
> * **It is worth stating that reviewer xSa6 pointed out our lack of ablation experiments, which was a misunderstanding.** As reviewer 2B2X commended, we conducted ablation experiments in the original paper to illustrate the contribution of each decoupled component.
>
> * In our refined work, we have added numerous ablation experiments to support the importance of our ideas, strategies, and framework.
>
> * As reviewer xSa6 mentioned that the presentation of this paper could be improved. We have improved the writing and we have reviewed and revised all our symbols, formulae and annotations to make them clearer and easier to understand.
>
> The above is a brief summary of our rebuttal, please see our response to reviewers or the Appendix F in revised paper for details.
>
> We have been working to provide intuitive, easily interpretable, theoretically supported and clearly motivated design ideas for GNN. We believe that our idea of decoupling structural, positional, and attribute information, as well as the decoupled design framework, both firstly proposed in the GT domain, are novel and relevant. I sincerely hope that the reviewers and area chair could read our rebuttal and discuss our work in the next phase. If we can still comment after the deadline, we will be more than happy to continue the discussion. Thanks in advance!
>
> Thanks again to all of you! Happy Thanksgiving!
>
> Best regards, Authors